# Socially driven negative feedback regulates activity and energy use in ant colonies

**Maurizio Porfiri** [1,2,3]*, **Nicole Abaid** [4], **Simon Garnier** [5]

1 Center for Urban Science and Progress, New York University Tandon School of Engineering, Brooklyn, New York, United States of America, 2 Department of Biomedical Engineering, New York University Tandon School of Engineering, Brooklyn, New York, United States of America, 3 Department of Mechanical and Aerospace Engineering, New York University Tandon School of Engineering, Brooklyn, New York, United States of America, 4 Department of Mathematics, Virginia Tech, Blacksburg, Virginia, United States of America, 5 Department of Biological Sciences, New Jersey Institute of Technology, Newark, New Jersey, United States of America

* mporfiri@nyu.edu

**Data Availability Statement:** All code used for numerical results is available on a GitHub repository at https://github.com/dynamicsystemslaboratory/Ant-Epidemic-Model.

## Abstract

Despite almost a century of research on energetics in biological systems, we still cannot explain energy regulation in social groups, like ant colonies. How do individuals regulate their collective activity without a centralized control system? What is the role of social interactions in distributing the workload amongst group members? And how does the group save energy by avoiding being constantly active? We offer new insight into these questions by studying an intuitive compartmental model, calibrated with and compared to data on ant colonies. The model describes a previously unexplored balance between positive and negative social feedback driven by individual activity: when activity levels are low, the presence of active individuals stimulates inactive individuals to start working; when activity levels are high, however, active individuals inhibit each other, effectively capping the proportion of active individuals at any one time. Through the analysis of the system's stability, we demonstrate that this balance results in energetic spending at the group level growing proportionally slower than the group size. Our finding is reminiscent of Kleiber's law of metabolic scaling in unitary organisms and highlights the critical role of social interactions in driving the collective energetic efficiency of group-living organisms.

## Author summary

Similarly to how larger organisms use less energy per unit mass than smaller ones, eusocial insects like ant colonies become more energy efficient as colony size increases. The mechanism underneath this efficiency remains a mystery. Here, we seek to uncover its origin in socially contagious "deactivation", which runs counter to more conventional ideas of excitatory social interactions. Beyond providing insight into the collective behavior of highly integrated social groups, our findings on activity regulation open the door to the design of engineered multi-agent systems, like robotic

**Funding:** This work was funded by the National Science Foundation under grant EF-2222418 awarded to M.P., N.A., and S.G. The funders had no role in study design, data collection and analysis, decision to publish, or preparation of the manuscript.

**Competing interests:** The authors have declared that no competing interests exist.

swarms or active matter, which may achieve efficient performance in both function and energy use.

## Introduction

As a complex social system—such as an ant colony or a human organization—increases in size, so does the need to better regulate the activities of its members [1]. Without regulation, individuals would randomly distribute their effort across time, wasting energy if too many individuals were active when the global workload does not require it, or wasting opportunities if too few individuals were to respond to increasing needs in the population. The question of activity—and hence energy—regulation is not unique to complex social systems; it is also increasingly studied in the context of artificial distributed systems such as fleets of robotic devices [2, 3] and Internet-of-Things networks [4], where redundancies are frequent and energy-wasting.

Both in natural and artificial distributed systems, the control of activities is often fully decentralized, making it impossible to consider global solutions for optimizing energy consumption. Studies on social insect colonies have shown that living systems can be surprisingly efficient in managing their collective activities [5]. In particular, their energy use per unit of mass appears to scale hypometrically with their colony size [6, 7]. In other words, larger colonies are more energy-efficient relative to their size than smaller ones, a phenomenon akin to Kleiber's law [8] that states that the rate of energy use by a biological system scales hypometrically with its size. Understanding how ant colonies—a fully decentralized system—regulate their activities to achieve energy efficiency could, therefore, have important repercussions for the design of human organizations and artificial distributed systems.

In the literature, activity regulation in social insects is typically divided between activation mechanisms that boost the number of individuals engaged in work, and inactivation mechanisms that reduce it [9, 10]. On the activation side, two types of mechanisms are usually invoked. First, individuals may sense workload-associated stimuli and, if their intensity exceeds a certain value (called the "response threshold"), start performing work to reduce this stimulation [11]. Such response thresholds have been found in several social insect species, for instance, for triggering a defense reaction to threats [12] or a foraging response to the presence of nutrients [13, 14].

Second, individuals already engaged in work can stimulate others to work, for instance, when the workload has outgrown their capacity [15]. This is frequently observed in social insects in the context of foraging. For example, ant workers that have found a resource will use a combination of chemical and tactile signals to stimulate other workers to join them in exploiting it [16]. Likewise, honeybee scouts that have found a new food patch execute a stereotypical "waggle dance" back at their colony that stimulates other bees to leave the nest and encodes the direction and distance to the resource [17]. Such recruitment processes are akin to a form of "social contagion", whereby the state of being active spreads in the group through local social interactions. As a result of this positive feedback loop, social organisms can mobilize a large portion of the available workforce quickly, facilitating rapid monopolization of resources [18] or overwhelming attackers by swiftly assembling defense forces around them [19].

While activation mechanisms are well studied and supported by documented examples, inactivation mechanisms are overlooked—especially those mediated by social interactions [20]. In existing models of activity regulation, inactivation is often treated as an intrinsic property of the individual rather than a socially driven one. For instance, inactivation has been

modeled as a workload-associated threshold [21], a limit on the time an individual can be active [22, 23], or a constant probability per unit time [9, 11]. In each of these cases, the social environment does not influence the duration of the activity of an individual, or, at best, does it very indirectly (for example, through the impact other individuals have on the quantity of work remaining to be done). Yet, socially driven inactivation mechanisms have been shown to play an important role in counterbalancing activation mechanisms in social insects. For instance, honeybees use inhibitory signals to slow down the recruitment of foragers when food storing at the nest cannot match the influx of new nectar and pollen [24] and to delay the maturation of hive workers into foragers when the forager population is already high [25]. In ants, crowding along a foraging trail can inhibit the deposition of trail pheromone, reducing the risk of traffic jam [26]. Repellent pheromone can also be used to discourage foragers from visiting unrewarding routes [27], for instance because the resource they lead to is now depleted or overcrowded.

In a recent work [28], we investigated the impact of a socially driven inactivation mechanism on the scaling of energy use in eusocial organisms. We proposed an explanation for Kleiber's law [8] in the context of colonies of harvester ants (*Pogonomyrmex californicus*) using data from Waters *et al.* [29]. Our explanation was based on scaling arguments adapted from urban science [30] and relied on a key biological phenomenon that we called "reverse social contagion"—a mechanism by which an individual engaged in a given behavior becomes more likely to interrupt this behavior as it interacts with more neighbors also engaged in the same behavior [31], see Fig 1a. Reverse social contagion can be observed, for instance, in the form of stop-signaling mechanisms [32] and blocking interactions [33], and has been proposed as a factor regulating collective decision-making [34, 35] and energy spending in social systems [28, 36].

In that study, we focused on reverse social contagion in the context of movement, where an individual would cease to move (inactivate) in response to a social environment where many neighbors are also moving (being active). Each colony was composed of $N$ individuals, of which $A$ were active. The individuals in the colony interacted through a network with $E \propto N^{3/2}$ links, as estimated from the data. To explain activity regulation in the form of a reduction in the fraction of active individuals as the colony grows, we proposed a balance between reverse social contagion and spontaneous social activation. The former should scale with $A^2/N^{1/2}$, given that the number of interactions of active individuals with other active individuals is estimated as the product between the total count of interactions in the colony, $E$, and the probability that two individuals are simultaneously active, $(A/N)^2$. The latter, instead, should scale with $N$, being an inherent property of the individuals. Balancing reverse social contagion and spontaneous social activation, one predicts a hypometric scaling of the colony's activity with respect to its size, $A \propto N^{3/4}$, akin to Kleiber's law [8].

In this article, we draw inspiration from this simple scaling argument to formulate a dynamic model for activity regulation in eusocial systems. The state-of-the-art modeling of the dynamics of eusocial systems has, so far, largely focused on socially driven positive feedback mechanisms to predict rhythmic activation patterns [22, 37, 38], ignoring the impact of social information on inactivation patterns. Here, instead, we explicitly include socially driven negative feedback through reverse social contagion. We present a detailed study of the model, in the form of a system of coupled differential equations and stochastic Monte Carlo simulations.

## Results

### State transitions

We consider a colony of size $N$. Each individual in the colony can be in one of three possible states: active (A), inactive (I), or refractory (R). Individuals in inactive or refractory states are

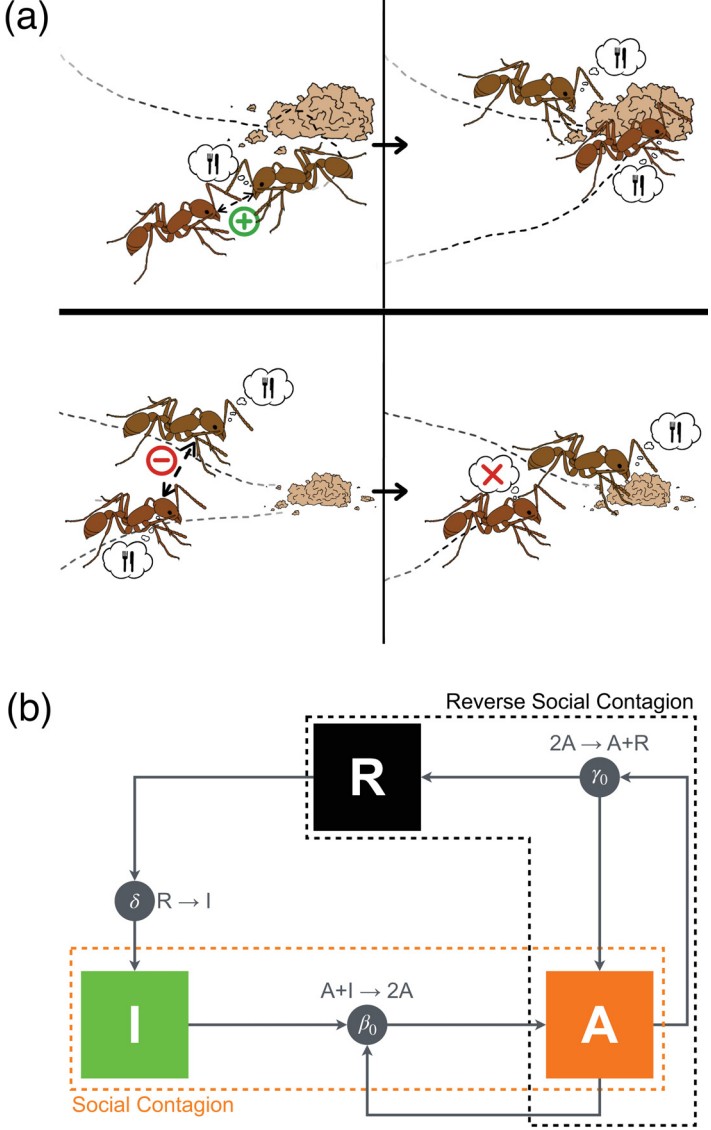

**Fig 1.** (a) Illustration of the concepts of social contagion (top) and reverse social contagion (bottom). (top) An inactive ant interacts with an ant engaged in a foraging task: through social contagion (for example, caused by active recruitment), it also begins foraging. (bottom) Two ants engaged in foraging interact: through reverse social contagion (for example, caused by steric exclusion), one of them ceases their activity to become inactive. *Image courtesy of Isabella Muratore, reprinted from Porfiri et al.* [28]. (b) State transition diagram for three classes of individuals (A, active, I, inactive, and R, refractory). The diagram describes two competing social feedback mechanisms (social contagion and reverse social contagion), along with a spontaneous transition from refractory to inactive state (that is, the completion of rest). Social contagion and completion of rest are well established in the literature since the seminal work by Goss and Deneubourg [9] (therein, inactive ants are called "activable inactives" and refractory ants "inactives" to specify that only the former ones can be activated). Reverse social contagion has been, instead, overlooked so far, thereby hindering our understanding of energy regulation in social groups.

practically indistinguishable (that is, they do not display any activity); however, only the inactive ones can be activated by social connections with active ants, whereas the refractory ones cannot.

We contemplate three coexisting phenomena, as shown in Fig 1b. First, an inactive individual can become active by interacting with an active individual (social contagion): $A + I \xrightarrow{\beta_0} 2A$,

where the transition rate $\beta_0 > 0$ is the probability per unit time that an inactive individual interacting with an active individual becomes active. Social contagion provides positive feedback, or autocatalysis, to the system [9]. Second, a refractory individual can spontaneously become inactive (hence, be ready for activation through social contagion): $R \xrightarrow{\delta} I$, where the transition rate $\delta > 0$ is the probability per unit time that a refractory individual spontaneously becomes inactive. This phenomenon, in probabilistic terms, is similar to a variable rest period after activity. Third, we hypothesize that an active individual can become refractory upon interaction with another active individual (reverse social contagion): $2A \xrightarrow{\gamma_0} R + A$, where the transition rate $\gamma_0 > 0$ is the probability per unit time that an active individual interacting with another active ant becomes refractory. This phenomenon creates negative feedback in the system, countering autocatalysis.

With respect to the autocatalytic ant colony model by Goss and Deneubourg [9] and the related mathematical model by da Silveira and Fontanari [38], we include reverse social contagion—an original contribution of this work—while making two simplifications. First, we neglect spontaneous activation and inactivation $I \to A$ and $A \to R$, respectively. Second, we do not explicitly consider a fixed rest period in which individuals must persist once entering the refractory state, before being ready to be activated. These simplifications are intended to reduce the number of model parameters and focus on a minimalistic model that could highlight the role of reverse social contagion on activity regulation.

## Compartmental model

We describe the colony dynamics through a compartmental model of the following form:

$$\dot{A}(t) = \frac{\beta_0 \langle k \rangle}{N} A(t) I(t) - \frac{\gamma_0 \langle k \rangle}{N} A^2(t), \tag{1a}$$

$$\dot{I}(t) = \delta R(t) - \frac{\beta_0 \langle k \rangle}{N} A(t) I(t), \tag{1b}$$

$$\dot{R}(t) = -\delta R(t) + \frac{\gamma_0 \langle k \rangle}{N} A^2(t). \tag{1c}$$

Here, $A(t)$, $I(t)$, and $R(t)$ are the number of active, inactive, and refractory individuals at time $t$, respectively, and $\langle k \rangle$ is the average degree of the network of interactions of the colony, encapsulating the number of interactions of each individual with other individuals at any time. Equation set (1) is similar to an epidemiological SIRS model, where recovery yields temporary immunity and recovered individuals return to the susceptible class at a given rate [39]. However, in contrast with epidemiological compartmental models where nonlinearities are typically restricted to mixed terms, we include a negative nonlinear quadratic term that captures reverse social contagion.

We hypothesize that the average degree scales with the network size as $\langle k \rangle = 2E_0 N^{\alpha-1}$ where $E_0 > 0$ is some scaling coefficient and $\alpha \in [1, 2]$ is the scaling exponent for the number of interactions with the colony size. A value close to 1 represents a network with a constant degree that does not vary with the colony size, and a value approaching 2 captures all-to-all interactions of a complete graph. Our previous observations on harvester ants yielded a value of $\alpha \simeq 3/2$ for physical proximity, as well as antennal contact networks [28]. Interestingly, the same value is also found in simulations of the classical Vicsek model for self-propelled particles when examining the dependence between the average degree in a particle cluster and its size [40]. We note that hypermetric scalings of the number of interactions with the group size have

been documented for different social interaction types, across several other classes than insects, including mammals (non-primates, primates, and humans), ray-finned fishes, birds, and reptiles. The lowest values for $\alpha$ have been recorded for online social friendship (humans), and the largest ones for physical contact (primates, birds, and insects) and grooming (primates) [41].

By construction, the sum of the three populations ($A(t)$, $I(t)$, and $R(t)$) is equal to the colony size, $N$, for all times; in fact, summing Eqs (1a)–(1c), we recover that $\dot{A}(t) + \dot{I}(t) + \dot{R}(t) = 0$. Next, we can show that given well-defined initial conditions for the three variables ($A_0$, $I_0$, and $R_0$ non-negative and summing to $N$), they all remain non-negative—see Materials and methods, "Positivity of the compartmental model."

Solving the system of algebraic equations given by $\dot{A} = \dot{I} = \dot{R} = 0$, we find two equilibria for the model, one trivial and one nontrivial—see equation set (4) from Materials and Methods, "Equilibria of the compartmental model." Through the study of the Jacobian of equation set (1) in correspondence of the two equilibria, we show that the trivial equilibrium is unstable and the other equilibrium is (locally and marginally) stable—see Materials and methods, "Local stability analysis of the non-trivial equilibrium." Just like endemic states in epidemic models, we can prove that the stability of the non-trivial equilibrium is global [42, 43]—see Materials and methods, "Global stability analysis of the non-trivial equilibrium." As a result, the model does not admit limit cycles, and any dynamics will ultimately converge to the non-trivial equilibrium.

The stable equilibrium is energetically favorable, whereby any network of interactions for which the average degree increases with the colony size ($\alpha > 1$) supports energetic regulation in the colony. In fact, the fractions of active and inactive individuals, $A^\star/N$ and $I^\star/N$, become smaller and smaller as the colony increases. The majority of the colony will be in a refractory state, which is the least energy-costly state for ants [44] and other social insects [45, 46]. For example, experiments on freely moving *Camponatus* ants by Lipp *et al.* [44] determined that the metabolic rate of walking ants increases four- to seven-fold over resting rates.

From basic calculus, one can establish that the number of active individuals increases with $\delta$ and decreases with $\gamma_0$, which indicates that slower rates of spontaneous refractory-to-inactive transition and faster rates of reverse social contagion facilitate energy savings by the colony. Likewise, increasing $\beta_0$ reduces the fraction of individuals in the inactive state, and, to a lesser extent, increases the fraction of active individuals.

## Scaling

For $\alpha = 1$, the average degree is independent of the colony size so that the populations of the non-trivial equilibrium in equation set (4) from Materials and Methods, "Equilibria of the compartmental model," scale isometrically with $N$, that is, $A^\star, I^\star, R^\star \propto N$. Interesting allometries that are reminiscent of Kleiber's law emerge for $\alpha > 1$, whereby taking the limit of equation set (4) from Materials and Methods, "Equilibria of the compartmental model," for $N \gg 1$ with $\langle k \rangle = 2E_0 N^{\alpha-1}$ yields

$$A^\star \simeq \sqrt{\frac{\delta}{2\gamma_0 E_0}} N^{\frac{3-\alpha}{2}}, \tag{2a}$$

$$I^\star \simeq \frac{1}{\beta_0} \sqrt{\frac{\delta\gamma_0}{2E_0}} N^{\frac{3-\alpha}{2}}, \tag{2b}$$

$$R^\star \simeq N. \tag{2c}$$

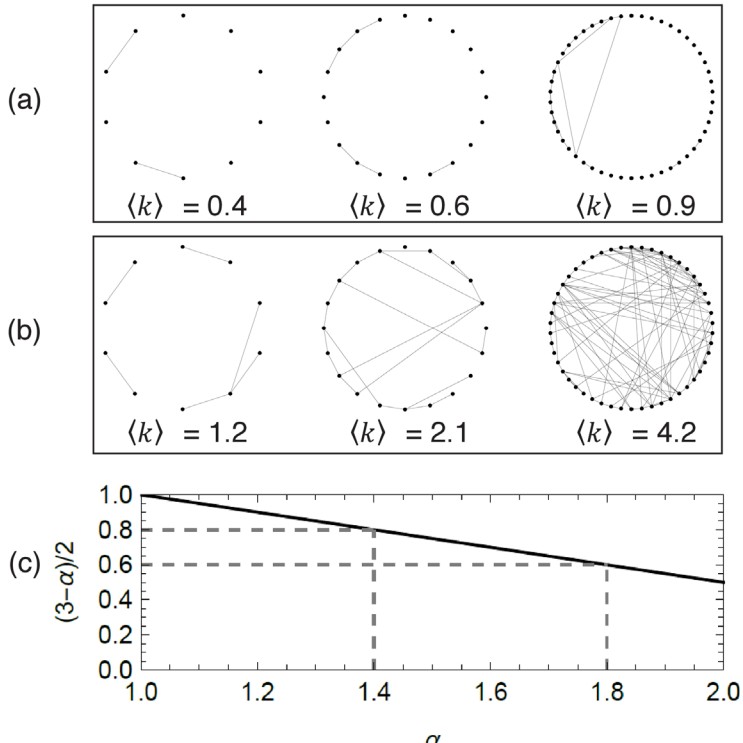

**Fig 2. Role of network connectivity on hypometric scaling of the collective activity.** (a,b) Sample networks with different number of nodes ($N = 10$, 20, and 50), for the same value of $E_0 = 0.0944$ from Porfiri *et al.* [28] and different values of $\alpha = 1.4$ (a) and $\alpha = 1.8$ (b), respectively. (c) Scaling exponent $(3 - \alpha)/2$ as a function of $\alpha$, illustrating activity regulation with increased connectivity: for $\alpha = 1.4$, increasing the network size contributes to a modestly sublinear growth of the number of active ants ($(3 - \alpha)/2 = 0.8$), while for $\alpha = 1.8$, increasing the network size yields dramatic sublinear growth of the number of active ants ($(3 - \alpha)/2 = 0.6$).

Fig 2 demonstrates how network connectivity impacts the scaling exponent above as network size varies. In the case $\alpha \gtrsim 3/2$—corresponding to spatial proximity and physical contact in harvester ants and other insects [41]—the number of active ants scales hypometrically with $N$ with an exponent ranging from approximately 1/2 to 3/4. Using the crude approximation that the energy cost for a colony is proportional to the number of active individuals, the observed scaling is in agreement with existing experimental observations of hypometric scaling of whole-colony metabolic rate with colony mass in ants [6, 29, 47, 48] and other group-living organisms [7, 49]. Since it is impossible to directly observe when an individual is in the refractory state, it is natural to ask whether this state is required for the model to show the scaling above. When the refractory state is excluded, hypometric scaling of the number of ants becomes unfeasible, demonstrating the need to force individuals to rest before being reactivated through a refractory state—see Materials and methods, "Compartmental model without refractory state."

In the limit $\beta_0 \to 0$, there is no mechanism for individuals to be activated and the non-trivial equilibrium converges to the trivial equilibrium. In this case, activity regulation does not emerge, pointing to the need for a balance between positive and negative social feedback. Notably, should one replace reverse social contagion with a spontaneous transition from active to refractory, any nonlinear allometry would disappear—see Materials and methods, "Compartmental model without reverse social contagion." The possibility of recovering Kleiber's

law of metabolic scaling in eusocial organisms relies on the balance between social contagion and reverse social contagion.

From a practical point of view, for equation set (2) to accurately approximate equation set (4) from Materials and Methods, "Equilibria of the compartmental model," one should have that $\frac{4\gamma_0\langle k\rangle}{\delta} \gg \left(1 + \frac{\gamma_0}{\beta_0}\right)^2$. For colonies of finite size, this requires $\beta_0$ to be sufficiently large compared to $\gamma_0$, that is, positive feedback (social contagion) not to be dominated by negative feedback (reverse social contagion); a caveat to this argument would be the case of slow spontaneous transition from refractory to inactive, for which $\delta$ could be very small.

## Model predictions for parameters calibrated on ant colonies

Model parameters are chosen as $E_0 = 0.0944$, $\alpha = 1.47$, $\gamma_0 = 1.21$ s$^{-1}$, and $\delta = 0.63$ s$^{-1}$, based on experimental results in Waters *et al.* [29], which we recently re-analyzed in the context of scaling [28]—see Materials and methods, "Calibration of model parameters on experimental data." We explore three different values of $\beta_0$ (namely, $\gamma_0/10$, $\gamma_0$ and $10\gamma_0$) to detail the interplay between the positive and negative feedback mechanisms of social and reverse social contagion, respectively.

We study the time-evolution of model in equation set (1) with initial conditions $A_0 = N - 1$, $I_0 = 1$, and $R_0 = 0$ to visualize the effect of reverse social contagion in bringing the activity of the colony towards equilibrium. Numerical results for $N = 500$ in Fig 3 confirm the global stability of the non-trivial equilibrium in equation set (4) from Materials and Methods, "Equilibria of the compartmental model," whereby the variables converge to the equilibrium even from initial conditions that are far away from the equilibrium. Changing the value of $\beta_0$ with respect to the other parameters affects the time scale of the evolution, as well as the organization of the colony. Specifically, increasing the value of $\beta_0$ accelerates convergence towards the steady state, due to stronger social contagion that facilitates the activation of individuals in the inactive status. Stronger positive feedback is also responsible for a reduced number of inactive individuals, as well as an increased number of active and refractory individuals, as one would predict from equation set (4) from Materials and Methods, "Equilibria of the compartmental model".

Asymptotic results in Eqs (2a) and (2b) for the number of active and inactive individuals at the steady state are close to exact values for the largest value of $\beta_0$, suggesting that scaling arguments could be safely used for order-of-magnitude calculations in the case of strong social contagion, see Fig 4. For smaller values of $\beta_0$, the agreement between asymptotic predictions and exact values is qualitatively equivalent, albeit asymptotic values should only be used as an order of magnitude estimation. In fact, predictions of the scaling exponents for all salient variables, identified from observations over a range of colony sizes from 10 to 1,000, indicate close agreement between equation set (2) and exact values—see Text A and Fig A in the S1 File.

## Monte Carlo simulations

In agreement with evidence on classical epidemiological models [50], our compartmental model is a close representation of the macroscopic response of the colony for a homogeneous network of interactions. We demonstrate this point using Monte Carlo simulations, a typical modeling strategy to study collective behavior in ants [37].

The compartmental model in equation set (1) could be viewed as a mean-field approximation of an agent-based model, in which ants interact over a temporal network following the state transitions in Results, "State transitions." More specifically, we simulate a network of $N$

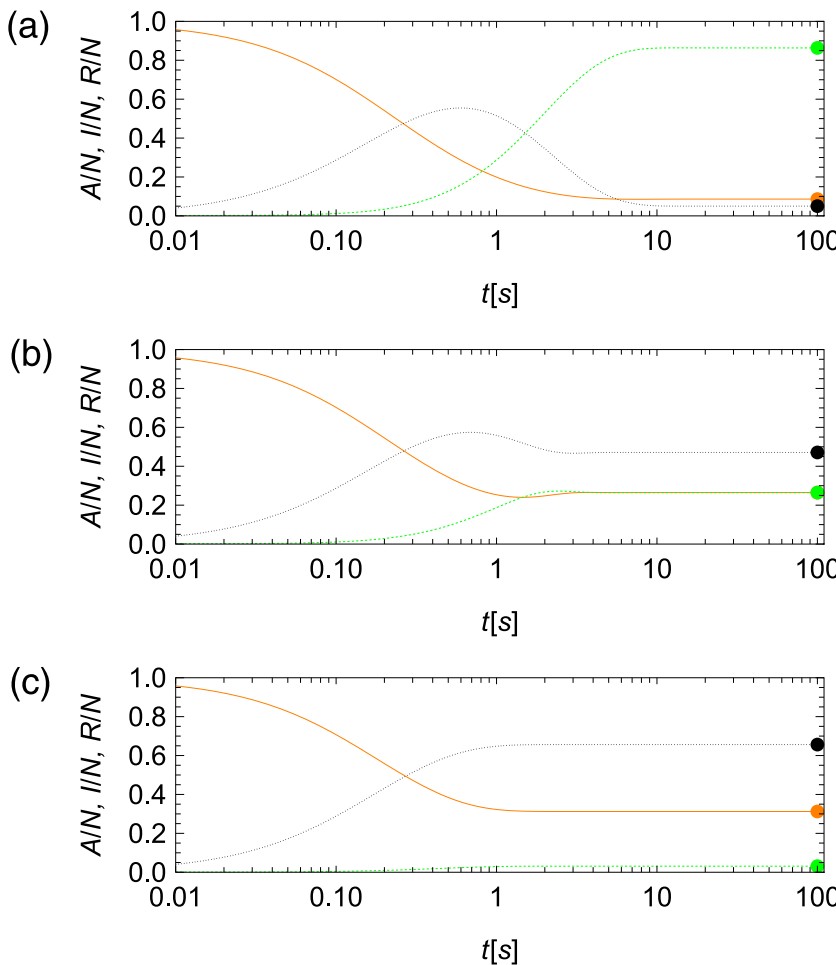

**Fig 3. Numerical predictions from compartmental model in equation set (1) for the time-evolution of the fraction of active ($A$, orange solid), inactive ($I$, green dashed), and refractory ($R$, black dotted) ants for a colony of $N = 500$ individuals and model parameters inspired by harvester ants plotted with time on a logarithmic scale.** (a) Reverse social contagion dominating social contagion ($\beta_0/\gamma_0 = 1/10$). (b) Reverse social contagion and social contagion at the same level ($\beta_0/\gamma_0 = 1$). (c) Social contagion dominating reverse social contagion ($\beta_0/\gamma_0 = 10$). Steady-state predictions in equation set (4) from Materials and Methods, "Equilibria of the compartmental model," are marked as colored dots in each panel.

nodes. Each node can be in state A, I, or R. The system evolves in discrete time according to the state transitions described in Results, "State transitions," with a time-step $\Delta t = 0.0667$ s. The network of interactions switches in time without memory, simulating complete mixing in the colony where any individual can in principle interact with any other ant. At each time step the network of interactions is drawn from an Erdős-Rényi network [51] with link probability $p = \langle k \rangle / N$ so that its average degree is $\langle k \rangle$. The model does not consider full motion trajectories, such as the work by Puckett *et al.* [52] and Ni *et al.* [53], and should be viewed as a limit case for full mixing. Fig 5 illustrates two consecutive time steps in the simulation.

We simulate a colony of $N = 500$ individuals for $T = 150$ time steps starting from the same initial condition, consisting of all individuals being active except for one that is inactive, analogously to the compartmental model. Model parameters are the same as in the compartmental model ($\alpha = 1.47$, $\gamma_0 = 1.21$ s$^{-1}$, and $\delta = 0.63$ s$^{-1}$). We choose the smallest

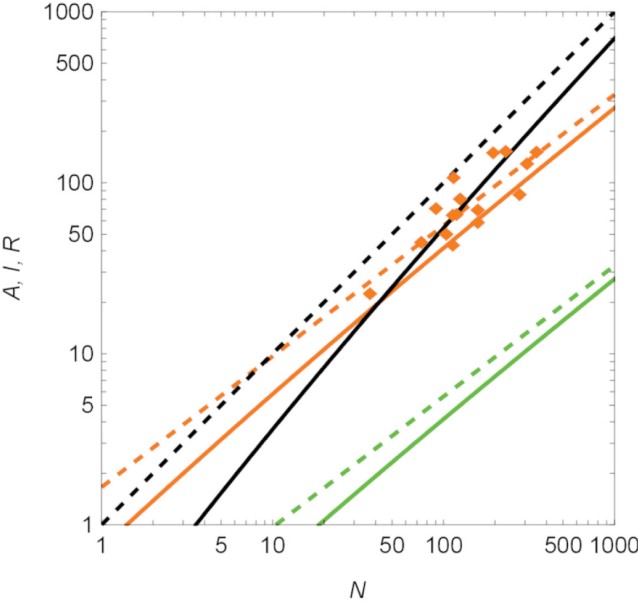

**Fig 4. Comparison between asymptotic predictions in equation set (2) (dashed lines) and exact values (solid lines) in equation set (4) from Materials and Methods, "Equilibria of the compartmental model," for the steady-state number of active ($A$, orange), inactive ($I$, green), and refractory ($R$, black) ants as a function of the colony size, $N$, for model parameters inspired by harvester ants and $\beta_0/\gamma_0 = 10$.** Experimental observations of harvester ants, presented by Porfiri *et al.* [28], are included for completeness (orange diamonds).

value of $\beta_0$ ($\beta_0 = \gamma_0/10$), corresponding to the slowest transient to best visualize the comparison between model predictions and agent-based simulations. We run 10 simulations and compute mean and standard deviations at each time step. As shown in Fig 6a, 6c and 6e, model predictions are in excellent agreement with the Monte Carlo simulations.

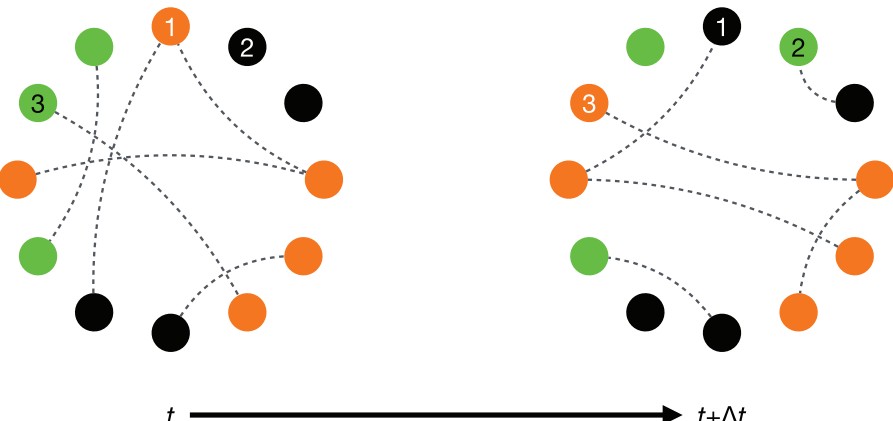

**Fig 5. Illustration of the Monte Carlo simulation, implemented on a switching Erdős-Rényi network.** Active, inactive, and refractory nodes are colored orange, green, and black, respectively. Example of reverse social contagion: node 1 is active at time $t$ and, through the interaction with another active node, it becomes refractory at $t + \delta t$. Example of spontaneous transition from refractory to inactive: at $t$, node 2 is refractory and it spontaneously become inactive at $t + \Delta t$. Example of social contagion: node 3 is inactive at $t$ and, through the interaction with an active node, it becomes active at $t + \Delta t$.

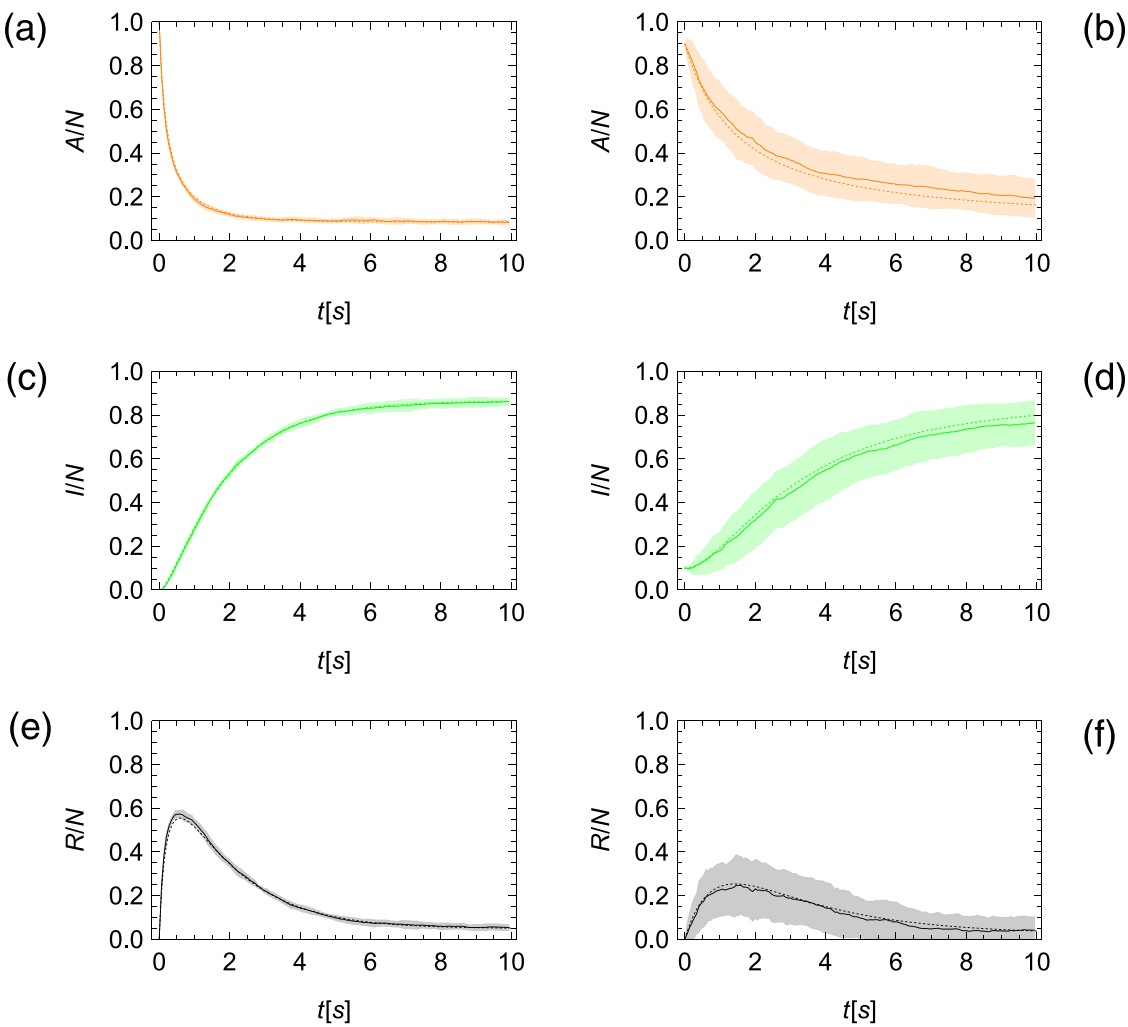

**Fig 6. Comparison between the numerical results of the compartmental model in equation set (1) (dashed line) and simulations of an agent-based model (solid line, mean, and dashed regions, one standard deviation above and below the mean) for a colony of *N* = 500 individuals (a,c,e) and *N* = 10 individuals (b,d,f).** (a,b) Fraction of active ants (*A*, orange). (c,d) Fraction of inactive ants (*I*, green). (e,f) Fraction of refractory ants (*R*, black).

The accuracy of the compartmental model is unaffected by reducing the colony size to *N* = 10, as shown in Fig 6b, 6d and 6f; therein, the number of realizations was raised to 100. Obviously, the variation among different realizations increases, but the mean is in excellent agreement with the compartmental model. Comparing results in the left and right columns of Fig 6, we confirm the hypometric scaling of the number of active individuals, whereby the fractions of active individuals increase more than twofold when lowering the size of the colony from 500 to 10.

## Discussion

We showed that efficient activity regulation can emerge in highly integrated social groups from a balance between two socially driven feedback processes. First, an autocatalytic process of social contagion results in the activation of inactive individuals, after stimulation by already active individuals. This positive feedback is then counterbalanced by an "autoinhibitory"

process that we call reverse social contagion where active individuals are increasingly likely to stop working as they interact with more active individuals. As a consequence, the number of active individuals in the group will be naturally capped at a proportion of the group size, whose value depends on the balance between the two aforementioned processes. This is compatible with observations of "lazy" individuals with very low levels of activity and commonly observed in large colonies of eusocial insects [54].

We also demonstrated that, when the number of interactions between the members of the group grows faster than the size of the group, the proportion of active individuals grows hypometrically with the group size. Hence, the group becomes energetically more efficient as its size increases. Both network and energy scalings are commonly observed in group-living organisms [6, 7, 41, 47], supporting the hypothesis that social interactions are critical in explaining the energetic properties of social groups (including human beings). Notably, this scaling does not occur when the refractory state is removed from the model, suggesting that the refractory state serving as a reservoir is critically important to recovering the relationship between activation and colony size observed in experimental data.

Finally, the reverse social contagion mechanism that we study here is fundamentally different from the inactivation mechanisms proposed in existing models of activity dynamics. Indeed, the latter are typically not socially driven, and they rely on constant duration of activity [22], constant rate of inactivation [9, 11], or the ability of an individual to estimate the amount of remaining work [21]. In the first two cases, activity regulation is entirely dependent on the activation process and the hypometric energy scaling disappears. In the last case, the inactivation process can depend on the social environment as in our model, albeit indirectly via stigmergic communication since the amount of remaining work is impacted by the actions of the active individuals. This scenario is already included in our model since it does not require specifying the mode of interaction (direct or indirect) between active individuals but simply its rate.

In conclusion, our study provides a generalizable explanation for activity regulation in social groups. Its predictions are in agreement with qualitative and quantitative observations of general activity patterns and energetic scaling found in highly integrated social species, such as ant colonies. Our next step is to incorporate multiple activity types and mobility in the model, towards a framework for the study of more complex and specific patterns of emergent division of labor and their impact on energy use in animal and human groups.

## Methods and materials

### Positivity of the compartmental model

To prove that given well-defined initial conditions, all the populations remain non-negative, we start by bounding from below the right-hand side of Eq (1c) and integrating over time to establish $R(t) \geq R_0 \exp(-\delta t) \geq 0$ through the comparison principle [55]. Substituting the above bound for $R(t)$ into Eq (1b) and applying the comparison principle again, we can bound the right-hand side and integrate over time to obtain $I(t) \geq I_0 \exp\left(-\frac{\beta_0 \langle k \rangle}{N} \int_0^t A(\tau) \mathrm{d}\tau\right) \geq 0$. Eq (1a), can be written as $\dot{A}(t) = f(t)A(t)$, where $f(t) = \frac{\langle k \rangle}{N}(\beta_0 I(t) - \gamma_0 A(t))$, so that $A(t) = A_0 \exp(\int_0^t f(\tau) \mathrm{d}\tau) \geq 0$.

### Equilibria of the compartmental model

Equation set (1) has two feasible equilibria (denoted with a superscript star), which are determined by setting to zero the left-hand side of two of the three equations in the set and imposing $A^\star + I^\star + R^\star = N$. The first equilibrium is the trivial one $A^\star = R^\star = 0$ and $I^\star = N$, where the

entire population is inactive. The other equilibrium is the positive root of a quadratic equation, obtained by solving Eq (1a) for $I^\star$ and Eq (1c) for $R^\star$ (assuming the left-hand sides to be zero and setting $A^\star \neq 0$). Following these steps, we establish

$$\frac{\gamma_0 \langle k \rangle}{\delta N} (A^\star)^2 + \left(1 + \frac{\gamma_0}{\beta_0}\right) A^\star - N = 0. \tag{3}$$

By solving for $A^\star$ and recalling the steps to obtain this equation from Eqs (1a) and (1c), we identify the equilibrium:

$$A^\star = \frac{\delta N}{2\gamma_0 \langle k \rangle} \varphi\left(\frac{\gamma_0}{\beta_0}, \frac{4\gamma_0 \langle k \rangle}{\delta}\right), \tag{4a}$$

$$I^\star = \frac{\delta N}{2\beta_0 \langle k \rangle} \varphi\left(\frac{\gamma_0}{\beta_0}, \frac{4\gamma_0 \langle k \rangle}{\delta}\right), \tag{4b}$$

$$R^\star = \frac{\delta N}{4\gamma_0 \langle k \rangle} \varphi^2\left(\frac{\gamma_0}{\beta_0}, \frac{4\gamma_0 \langle k \rangle}{\delta}\right), \tag{4c}$$

where we introduce the function

$$\varphi(x, y) = \sqrt{(1 + x)^2 + y} - (1 + x). \tag{5}$$

Note that, in principle, Eq (3) has two solutions, but the other is negative, since the model parameters are all positive.

## Local stability analysis of the non-trivial equilibrium

We determine the following expression for the Jacobian:

$$J^\star = \begin{bmatrix} \dfrac{\langle k \rangle (\beta_0 I^\star - 2\gamma_0 A^\star)}{N} & \dfrac{\beta_0 \langle k \rangle A^\star}{N} & 0 \\[2ex] -\dfrac{\beta_0 \langle k \rangle I^\star}{N} & -\dfrac{\beta_0 \langle k \rangle A^\star}{N} & \delta \\[2ex] \dfrac{2\gamma_0 \langle k \rangle A^\star}{N} & 0 & -\delta. \end{bmatrix} \tag{6}$$

By replacing the trivial equilibrium ($A^\star = R^\star = 0$ and $I^\star = N$), we obtain a block-diagonal matrix, whose eigenvalues are 0, $-\delta$, and $\beta_0 \langle k \rangle$; the latter is positive, hence the instability of the equilibrium.

To assess the stability of the non-trivial equilibrium in equation set (4), we apply Routh-Hurwitz criterion [56] to the characteristic polynomial of the Jacobian. Specifically, the characteristic polynomial of the Jacobian in correspondence to the non-trivial equilibrium can be factored as

$$p(s) = -s(a_0 s^2 + a_1 s + a_2) \tag{7}$$

where

$$a_0 = 1, \tag{14}$$

$$a_1 = \frac{\sqrt{\delta}(\beta_0 + \gamma_0)}{2\beta_0\gamma_0}\sqrt{4\beta_0^2\gamma_0\langle k \rangle + \delta(\beta_0 + \gamma_0)^2} - \frac{\delta(\beta_0^2 + \gamma_0^2)}{2\beta_0\gamma_0}, \tag{8a}$$

$$a_2 = \frac{\delta}{2\beta_0\gamma_0}\left(4\beta_0^2\gamma_0\langle k \rangle + \delta(\beta_0 + \gamma_0)^2\right) - \frac{\delta\sqrt{\delta}(\beta_0 + \gamma_0)}{2\beta_0\gamma_0}\sqrt{4\beta_0^2\gamma_0\langle k \rangle + \delta(\beta_0 + \gamma_0)^2}. \tag{8b}$$

Coefficients $a_1$ and $a_2$ are positive, like $a_0$, as one might verify by comparing the squares of the two summands in each expression. Based on the Routh-Hurwitz stability criterion, the roots of the second-order polynomial in Eq (7) have strictly negative real parts, which yields marginal stability of the non-trivial equilibrium. The null eigenvalue is related to the original constraints that $A(t) + I(t) + R(t) = N$.

## Global stability analysis of the non-trivial equilibrium

To prove global stability of the non-trivial equilibrium, we replace $I(t)$ with $N - A(t) - R(t)$ into Eqs (1a) and (1c) and study global asymptotic convergence of the $A(t)$ and $R(t)$ towards $A^\star$ and $R^\star$ in Eqs (4a) and (4c). We construct a Lyapunov function for this two-dimensional system that yields global stability [55]. We take inspiration from Goh-Lotka-Volterra Lyapunov functions to cope with the nonlinearity in the system [43]. Specifically, we propose the following function:

$$V(A, R) = \frac{(R - R^\star)^2}{2N^2} + \frac{\gamma_0}{2N^2\beta_0}\left[A^2 - (A^\star)^2 - (A^\star)^2\ln\frac{A^2}{(A^\star)^2}\right]. \tag{9}$$

Such a function is positive definite in $[0, N]^2$, it is zero at $(A^\star, R^\star)$, and all its level sets are bounded. For $V$ to be a Lyapunov function, we must show that $\dot{V}(A, R) < 0$ for any $(A, R) \in [0, N]^2$, except for $(A^\star, R^\star)$, where $\dot{V}(A^\star, R^\star) = 0$.

To prove this claim, we compute the time derivative of $V(A, R)$ along the system trajectories, using equation sets (1) and (4) to obtain

$$\dot{V}(A, R) = -\frac{\delta}{N^2}(R - R^\star)^2 - \frac{\langle k \rangle}{N^3}\left(\gamma_0 + \frac{\gamma_0^2}{\beta_0}\right)(A - A^\star)\left[A^2 - (A^\star)^2\right]. \tag{10}$$

The first summand is a quadratic function in $R$ that is negative in $[0, N]$, except for $R = R^\star$, where it is zero. Likewise, the second summand is a cubic function in $A$ that is negative in $[0, N]$, except for $A = A^\star$ where it is zero. As a result, the claim follows.

## Compartmental model without refractory state

We study a variation of equation set (1) where the refractory state is removed. Within this variation, active ants become inactive and inactive ants become active, both through social

interaction with active individuals. The equation set becomes

$$\dot{A}(t) = \frac{\beta_0 \langle k \rangle}{N} A(t) I(t) - \frac{\gamma_0 \langle k \rangle}{N} A^2(t), \tag{11a}$$

$$\dot{I}(t) = -\frac{\beta_0 \langle k \rangle}{N} A(t) I(t) + \frac{\gamma_0 \langle k \rangle}{N} A^2(t). \tag{11b}$$

Analogously to the study of the system without reserve social contagion in equation set (12), we can find the two equilibria for this system: the trivial equilibrium $(A^\star, I^\star) = (0, N)$ and the endemic equilibrium $(A^\star, I^\star) = \left( \frac{\beta_0}{\beta_0 + \gamma_0} N, \frac{\gamma_0}{\beta_0 + \gamma_0} N \right)$. Hence, without refractory state, the number of active ants will scale isometrically with the colony size.

## Compartmental model without reverse social contagion

We study a variation of equation set (1) where reverse social contagion is replaced with a spontaneous transition from active to refractory at a constant rate $\mu > 0$,

$$\dot{A}(t) = \frac{\beta_0 \langle k \rangle}{N} A(t) I(t) - \mu A(t), \tag{12a}$$

$$\dot{I}(t) = \delta R(t) - \frac{\beta_0 \langle k \rangle}{N} A(t) I(t), \tag{12b}$$

$$\dot{R}(t) = -\delta R(t) + \mu A(t). \tag{12c}$$

Searching for equilibria of the system, we set the left-hand side of each equation to zero and solve for $A^\star$, $I^\star$, and $R^\star$. In addition to the trivial equilibrium presented in Results, "Compartmental model" ($A^\star = R^\star = 0$ and $I^\star = N$), we determine the following equilibrium for $\mu < \beta_0 \langle k \rangle$: $A^\star = N \frac{\delta}{\beta_0 \langle k \rangle} \frac{\beta_0 \langle k \rangle - \mu}{\delta + \mu}$, $I^\star = \frac{\mu}{\beta_0 \langle k \rangle} N$, and $R^\star = N \frac{\mu}{\beta_0 \langle k \rangle} \frac{\beta_0 \langle k \rangle - \mu}{\delta + \mu}$. Hence, without reverse social contagion, the number of active ants would scale isometrically with colony size, whereby $A^\star \simeq \frac{\delta}{\delta + \mu} N$ for $\alpha = 1$ and $N \gg 1$, and simply $A^\star = \frac{\delta}{2\beta_0 E_0} \frac{2\beta_0 E_0 - \mu}{\delta + \mu} N$ for $\alpha = 1$.

## Calibration of model parameters on experimental data

The experimental dataset consists of 16 manually tracked videos of harvester ant colonies in $248 \times 248$ mm colony nest enclosures [6]. The length of each video was 30 s and the resolution was $\Delta t = 66.7$ ms, corresponding to 15 frames per second—note that manual tracking was performed on downsampled videos, retaining one every five frames. The size of the colony was different in each video, ranging from $N = 40$ to 360 workers (nominal values).

Network parameters $E_0$ and $\alpha$ were estimated in our prior work [28] by constructing a spatial interaction network as follows. For each colony and each frame, we created an undirected interaction network, where two ants were connected by an edge if they were within a distance of 6 mm, corresponding to one body length. Then, we averaged the number of edges, $E$, in each video to compute the total number of edges for each colony. Finally, we fitted $E$ against $N$ in the logarithmic scale.

From our previous research [28], we could also estimate the ratio $\delta/\gamma_0$. Therein, for each pair of consecutive frames in a video, we identified active ants as those that would move at least one pixel. For each video, we computed the total number of active ants, $A$ (also reported in Fig 4). Then, we fitted the overall extent of the reverse social contagion, $2A^2 E/N^2$, against $N$

to determine a slope $q = 0.519$. By comparing against model predictions for $A^\star$ in Eq (2a), we estimate $\delta/\gamma_0 = 0.519$—such an estimate relies on the limit $N \gg 1$.

To tease out $\delta$ from $\gamma_0$, we separately estimated $\gamma_0$ through an additional analysis of the videos. Different from the previous analyses, this step required us to score the activity of each ant in time, rather than once for the entire video. In particular, at each frame, we considered an ant to be (instantaneously) active if it moved at least a pixel from the previous frame. Any ant that was not scored as active was deemed to be either inactive or in a refractory state, the two being indistinguishable from video data. Across videos, we counted: i) the total number of tracked frames in which two active ants interacted (9861), and ii) the total number of instances in which two active ants interacted in one of these frames and then one became not active in the next tracked frame (800). By dividing ii) by i), we estimated the probability of reverse social contagion (0.0811), which, upon dividing by $\Delta t$ yielded our estimation for $\gamma_0$ of 1.21 s$^{-1}$. Based on the value of $q$, we then estimated $\delta$ as 0.63 s$^{-1}$.

Estimating $\beta_0$ without being able to discern inactive from refractory ants was not feasible. In fact, an argument equivalent to the one used to estimate $\gamma_0$ could not be pursued for $\beta_0$. Given that only inactive ants could cause social contagion, the probability of an event like A + I or R $\to$ 2 A should be interpreted as the probability of social contagion ($\beta_0 \Delta t$) times the probability that an ant that was not active was actually inactive ($I^\star/(I^\star + R^\star)$ from equation set (2)). Such a probability is, however, independent of $\beta_0$. Alternative ways around this issue would require access to longer time series to study transient phenomena, or a larger set of experimental trials to retain the full dependence of the population variables on the colony size in equation set (4).

## Supporting information

**S1 File. Adequacy of the scaling formulas.**
(PDF)

## Acknowledgments

We are thankful to Ofek Lauber, Gian Carlo Maffettone, and Michael Napoli for verifying the derivations and to Eighdi Aung for helping with the video analysis.

## Author Contributions

**Conceptualization:** Maurizio Porfiri, Simon Garnier.

**Data curation:** Maurizio Porfiri.

**Formal analysis:** Maurizio Porfiri, Nicole Abaid.

**Funding acquisition:** Maurizio Porfiri, Nicole Abaid, Simon Garnier.

**Investigation:** Maurizio Porfiri, Nicole Abaid, Simon Garnier.

**Methodology:** Maurizio Porfiri.

**Project administration:** Maurizio Porfiri.

**Resources:** Maurizio Porfiri.

**Software:** Maurizio Porfiri.

**Supervision:** Maurizio Porfiri.

**Validation:** Maurizio Porfiri.

**Visualization:** Maurizio Porfiri, Nicole Abaid, Simon Garnier.

**Writing – original draft:** Maurizio Porfiri, Simon Garnier.

**Writing – review & editing:** Maurizio Porfiri, Nicole Abaid, Simon Garnier.

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
