## [Decision Letter · Decision Letter 0]

11 Oct 2024

Dear Dr. Porfiri,

Thank you very much for submitting your manuscript "Socially driven negative feedback regulates activity and energy use in ant colonies" for consideration at PLOS Computational Biology. As with all papers reviewed by the journal, your manuscript was reviewed by members of the editorial board and by several independent reviewers. The reviewers appreciated the attention to an important topic. Based on the reviews, we are likely to accept this manuscript for publication, providing that you modify the manuscript according to the review recommendations.

This is an interesting and well-motivated paper that examines the potential role of negative feedback signals in regulating activity levels in foraging social insect colonies, leading to a metabolic scaling-like law for increasing colony size. Previous work on negative feedback in foraging social insect colonies has proposed its role in signalling danger (Nieh and colleagues), lack of profitability (Robinson and colleagues), aiding decision dynamics (Seeley and colleagues), and suppressing variation in performance (Reina and Marshall). The present manuscript thus makes an interesting complementary contribution, albeit one slightly foreshadowed in the literature, as discussed below).

While the presentation is generally clear, reviewer 1 feels that the key scaling law argument could be presented graphically. While I feel that the authors' present figure 3 does this to a degree, I agree with the reviewer that it could be presented in other, more intuitive ways. I would also suggest that some of the detail of the derivations could be moved into supplementary material, since the primary audience will be biologists, and studies show that equations inversely correlate with citation rates in the life sciences (Fawcett and Higginson, PNAS 109 (29) 11735-11739).

In addition, I have some comments of my own on the relationship between the present manuscript and existing work in the field - the authors note the existence of the above prior work on negative signalling, but in proposing their new model I am not sure they fully appreciate its relationship to previous models, in which energetic considerations of negative feedback have explicitly been alluded to. The model of the present manuscript is essentially equivalent to the honeybee stop-signalling model of Seeley et al. with two changes - one is the collapse of a decision problem into a single nest site problem, the other is the introduction of an intermediate 'recovering' class for recipients of negative signalling, bringing the model closer to the epidemiological literature it has long paralleled. From pen and paper maths I didn't quickly convince myself that the introduction of the recovering class would qualitatively change the authors' conclusions on energetic scaling, and it might be interesting to also analyse the simpler model to see if this is the case. This would help highlight the relationship with previous work where, in the binary decision case of the honeybee model, in another paper the authors explicitly discussed the energetic impact observed for negative feedback (first paragraph of Discussion of Pais et al., cited), where they also discuss a possible trade-off. While the present study makes standalone contributions, these links to the previous literature would strengthen it, in my opinion. I also felt that other work on regulation of energetic expenditure by social insect colonies in the absence of negative feedback, for example the work of Deborah Gordon, could be cited.

Finally, it may interest the authors to note that they could substantially automate aspects of the derivation and analysis of systems such as their present model, as well as disseminate effectively, through the use of an open source tool (mum.readthedocs.io) - a brief notebook analysing their model is available from the Editor on request.

Sincerely,

James A.R. Marshall, BSc, PhD

Academic Editor

PLOS Computational Biology

Zhaolei Zhang

Section Editor

PLOS Computational Biology

This is an interesting and well-motivated paper that examines the potential role of negative feedback signals in regulating activity levels in foraging social insect colonies, leading to a metabolic scaling-like law for increasing colony size. Previous work on negative feedback in foraging social insect colonies has proposed its role in signalling danger (Nieh and colleagues), lack of profitability (Robinson and colleagues), aiding decision dynamics (Seeley and colleagues), and suppressing variation in performance (Reina and Marshall). The present manuscript thus makes an interesting complementary contribution, albeit one slightly foreshadowed in the literature, as discussed below).

While the presentation is generally clear, reviewer 1 feels that the key scaling law argument could be presented graphically. While I feel that the authors' present figure 3 does this to a degree, I agree with the reviewer that it could be presented in other, more intuitive ways. I would also suggest that some of the detail of the derivations could be moved into supplementary material, since the primary audience will be biologists, and studies show that equations inversely correlate with citation rates in the life sciences (Fawcett and Higginson, PNAS 109 (29) 11735-11739).

In addition, I have some comments of my own on the relationship between the present manuscript and existing work in the field - the authors note the existence of the above prior work on negative signalling, but in proposing their new model I am not sure they fully appreciate its relationship to previous models, in which energetic considerations of negative feedback have explicitly been alluded to. The model of the present manuscript is essentially equivalent to the honeybee stop-signalling model of Seeley et al. with two changes - one is the collapse of a decision problem into a single nest site problem, the other is the introduction of an intermediate 'recovering' class for recipients of negative signalling, bringing the model closer to the epidemiological literature it has long paralleled. From pen and paper maths I didn't quickly convince myself that the introduction of the recovering class would qualitatively change the authors' conclusions on energetic scaling, and it might be interesting to also analyse the simpler model to see if this is the case. This would help highlight the relationship with previous work where, in the binary decision case of the honeybee model, in another paper the authors explicitly discussed the energetic impact observed for negative feedback (first paragraph of Discussion of Pais et al., cited), where they also discuss a possible trade-off. While the present study makes standalone contributions, these links to the previous literature would strengthen it, in my opinion. I also felt that other work on regulation of energetic expenditure by social insect colonies in the absence of negative feedback, for example the work of Deborah Gordon, could be cited.

Finally, it may interest the authors to note that they could substantially automate aspects of the derivation and analysis of systems such as their present model, as well as disseminate effectively, through the use of an open source tool (mum.readthedocs.io) - a brief notebook analysing their model is available from the Editor on request.

Reviewer's Responses to Questions

**Comments to the Authors:**

Reviewer #1: The paper shows that modelling ant foraging dynamics with a simple model of information spreading that combines positive and negative feedback can give interesting results on system size scaling. The author’s model shows that having nonlinear (social) negative feedback leads to different dynamics than only having linear (nonsocial) negative feedback. They also discuss that it can be interesting to look at this scaling in terms of the colony's energetic consumption.

I agree with the authors that this problem is interesting, worth studying, and relatively understudied.

I imagine (from what I could understand) that the key point of the article is the hypometrical growth of active-vs-total group size. This is interesting and worth discussing, so I am in favour of the eventual publication of this article, however, I would like first to see the hypometrical growth results presented more clearly. Now this result is only (possibly) hidden in fig. 3 with the orange line slope lower than 1, or by eyeballing 5(a) and 5(b).

My suggestion (but authors can come up with better solutions too) is to show two figures:

(1) easy interpretation of comparing a set of scaling lines of A (vs N on x-axis) for selected relevant parameters

(2) full space exploration, e.g., through a heatmap of the scaling slope of A (i.e. a heatmap of the slopes of the orange lines of fig.3) for all the range of parameters changed on the two axes (possibly beta and alpha).

Authors indicate (l. 116) that “we neglect spontaneous activation and inactivation I -> A and

A -> R” in order to “reduce the number of model parameters and focus on a minimalistic model”.

Previous analyses of similar models of opinion dynamics have shown the importance of spontaneous (i.e. nonsocial) state change (e.g. noisy voter models, Herrerías-Azcue & Galla Phys. Rev. E 2019, Reina et al. Comm.Phys. 2023), leading to qualitative changes of the mean-field dynamics. Could it also be the case with this model?

The authors say (in the discussion) "inactivation mechanisms proposed in existing models of activity dynamics. Indeed, the latter are typically not socially driven".

Please also comment on models that included socially-driven inactivation systems, e.g. [26, 29], and other stop signalling models (e.g. Bidari et al. RS Open Science 2019) or negative pheromone studies (e.g. Elva Robinson’s work).

Simulation method. The authors use an Erdos Renyi network that they rewire each timestep, given that the topology is randomly changed at each timestep, what is the difference between using an ER net and the more widely spread approach of sampling one random neighbor from the full network? (i.e., random pairwise interaction, with state change triggered with state transition probabilities). If there is no reason for doing so, it looks like an unnecessary complication of the model which should be removed (as model simplicity is stated as one of the contributions). Otherwise, if there is a reason (e.g. ER allows you to study the impact of <k> and to appreciate different scaling on k with different levels of hypometric growth); then, this result should be shown as it’s very interesting.

Results issues:

* l. 227 - “with initial conditions A0 = N − 1, I0 = 1, and R0 = 0

in fig.2, at t=0, R0 is higher than I0. This is confusing, please revise.

* l. 424 (Text S1) - unknown what C_A C_I and C_R are. Please explain.

Presentation issues:

* fig 1a is identical to fig.1 of the previous PNAS Nexus article.

* The section "Compartmental models" can be reduced in length as the mathematical steps for ODE stability analysis should go in the methods sections and the results text could focus more on presenting the model and discussing stability results, eq. (3) and (5), rather than deriving them.

*l. 277 “top and bottom rows” -> “left and right columns”?</k>

Reviewer #2: The authors argue and demonstrate using results from a previous study on effect of reserve social contagion and stability analysis of a set of mathematical equations that the energy spending in ant colonies grows proportionally slower than the group size. The study offers a potential answer to how large social groups regulate their activities (or energy spending in this case) without a centralized control system and makes a valuable contribution in this field.

**Have the authors made all data and (if applicable) computational code underlying the findings in their manuscript fully available?**

Reviewer #1: None

Reviewer #2: **No: **available from the authors upon request

PLOS authors have the option to publish the peer review history of their article (what does this mean?). If published, this will include your full peer review and any attached files.

Reviewer #1: No

Reviewer #2: No

Figure Files:

Data Requirements:

Reproducibility:

References:

---

## [Editor Report · Decision Letter 1]

8 Nov 2024

Dear Dr. Porfiri,

We are pleased to inform you that your manuscript 'Socially driven negative feedback regulates activity and energy use in ant colonies' has been provisionally accepted for publication in PLOS Computational Biology.

Best regards,

James A.R. Marshall, BSc, PhD

Academic Editor

PLOS Computational Biology

Zhaolei Zhang

Section Editor

PLOS Computational Biology

Feilim Mac Gabhann

Editor-in-Chief

PLOS Computational Biology

Jason Papin

Editor-in-Chief

PLOS Computational Biology

Thanks for your work improving the exposition and referencing of your results. I am happy to recommend this for publication now.

---

## [Editor Report · Acceptance letter]

15 Nov 2024

PCOMPBIOL-D-24-01331R1 

Socially driven negative feedback regulates activity and energy use in ant colonies

Dear Dr Porfiri,

I am pleased to inform you that your manuscript has been formally accepted for publication in PLOS Computational Biology. Your manuscript is now with our production department and you will be notified of the publication date in due course.

With kind regards,

Marianna Bach
